# Malnutrition According to GLIM Criteria Is Associated with Mortality and Hospitalizations in Rehabilitation Patients with Stable Chronic Obstructive Pulmonary Disease

**DOI:** 10.3390/nu13020369

**Published:** 2021-01-26

**Authors:** Vanesa Dávalos-Yerovi, Ester Marco, Dolores Sánchez-Rodríguez, Xavier Duran, Delky Meza-Valderrama, Diego A. Rodríguez, Elena Muñoz, Marta Tejero-Sánchez, Maria Dolors Muns, Anna Guillén-Solà, Esther Duarte

**Affiliations:** 1Physical Medicine and Rehabilitation Department, Parc Salut Mar (Hospital del Mar Hospital de l’Esperança), Passeig Maritim de la Barceloneta 25-29, 08003 Barcelona, Spain; ndavalos@parcdesalutmar.cat (V.D.-Y.); mtejero@psmar.cat (M.T.-S.); aguillen@psmar.cat (A.G.-S.); eduarte@psmar.cat (E.D.); 2Rehabilitation Research Group, Institut Hospital del Mar d’Investigacions Mèdiques (IMIM), Carrer del Doctor Aiguader 88, 08003 Barcelona, Spain; MDCSanchezRodriguez@parcdesalutmar.cat (D.S.-R.); dmezaconcepcion@psmar.cat (D.M.-V.); elenamunozredondo@gmail.com (E.M.); mmuns@psmar.cat (M.D.M.); 3School of Medicine, Universitat Autònoma de Barcelona (UAB), Campus Universitari Mar, Carrer del Doctor Aiguader 80, 08003 Barcelona, Spain; darodriguez@psmar.cat; 4School of Medicine, Universitat Internacional de Catalunya, Carrer Josep Trueta s/n, Sant Cugat del Vallès, 08195 Barcelona, Spain; 5Geriatrics Department, Centre Fòrum-Hospital del Mar, Carrer Llull 410, 08019 Barcelona, Spain; 6Department of Health Sciences (CEXS), Universitat Pompeu i Fabra, Carrer del Doctor Aiguader 80, 08003 Barcelona, Spain; 7Methodology and Biostatistics Support Unit, Institut Hospital del Mar d’Investigacions Mèdiques (IMIM), Carrer del Doctor Aiguader 88, 08003 Barcelona, Spain; xduran@imim.es; 8Physical Medicine and Rehabilitation Department, National Institute of Physical Medicine and Rehabilitation (INMFER), Panama City 0819, Panama; 9Physical Medicine and Rehabilitation Department, Caja de Seguro Social (C.S.S.), Panama City 0824, Panama; 10Respiratory Medicine Department, Hospital del Mar, Passeig Maritim de la Barceloneta 25-29, 08003 Barcelona, Spain; 11Centro de Investigación en Red de Enfermedades Respiratorias (CIBERES), Instituto de Salud Carlos III (ISCIII), Avenida Monforte de Lemos 3-5, Pabellón 11, 28019 Madrid, Spain; 12Endocrinology and Nutrition Department, Hospital del Mar-Parc de Salut Mar. Passeig Marítim de la Barceloneta, 25, 29, 08003 Barcelona, Spain

**Keywords:** malnutrition, GLIM, mortality, hospitalization, chronic pulmonary obstructive disease, rehabilitation

## Abstract

Malnutrition has a negative impact on patients with chronic pulmonary obstructive disease (COPD). The purpose of this study was to assess the prevalence of malnutrition, defined by the Global Leadership Initiative for Malnutrition (GLIM), in stable COPD patients referred to pulmonary rehabilitation, and to explore potential associations of malnutrition according to GLIM, and its components, with increased risk of mortality and hospitalizations in 2 years. In a post-hoc analysis of a prospective cohort of 200 rehabilitation patients with stable COPD, main outcome variables were hospital admissions, length of stay, and mortality during a 2-year follow-up. Covariates were malnutrition according to GLIM and its phenotypic criteria: unintentional weight loss, low body mass index (BMI), and low fat-free mass (FFM). Univariate and multivariate analysis were performed using logistic and proportional hazard Cox regression. Malnutrition according to GLIM showed 45% prevalence and was associated with increased mortality risk. Low age-related BMI and FFM were independently associated with mortality, which persisted after adjustment for age and lung function. Malnutrition and low BMI were also associated with increased risk of hospitalization. Malnutrition according to GLIM criteria was highly prevalent in rehabilitation patients with COPD and was associated with nearly 3 times greater mortality and hospitalization risk.

## 1. Introduction

Chronic obstructive pulmonary disease (COPD), a heterogeneous disease with pulmonary and extrapulmonary manifestations, is the third-highest cause of mortality worldwide, claiming 3 million lives in 2016 [1]. It is also highly associated with morbidity and disability [2] and frequently associated with concomitant diseases, including nutritional disorders [3]. Malnutrition is associated with adverse medical consequences contributing to frailty, morbidity, and mortality [4,5,6,7,8]. The prevalence of malnutrition in patients with COPD ranges from 20% to 45%, depending on the setting and the diagnostic methods used [9,10,11]. It often remains underdiagnosed and undermanaged despite a wide range of effective therapeutic approaches in this population [12,13].

In 2015, the European Society of Clinical Nutrition and Metabolism (ESPEN) proposed a consensus statement on nutrition for all age ranges and healthcare settings, thereby providing a unified, simple, and reliable tool for malnutrition diagnosis, independently of etiology [14], which was followed by European guidelines on definitions and terminology of clinical nutrition [15]. Given the lack of a worldwide consensus on diagnostic criteria to be applied in all clinical settings and the new evidence supporting the influence of disease and inflammation on malnutrition, the Global Leadership Initiative on Malnutrition (GLIM) engaged the majority of nutrition societies in an effort to standardize the diagnosis of malnutrition in clinical settings [16]. The GLIM definition is a three-step approach: patients are first identified by any validated screening tool, and then diagnosed if they fulfill at least one phenotypical characteristic (non-volitional weight loss, low body mass index (BMI), reduced muscle mass) and one etiological criterion (reduced food intake or assimilation, disease burden, inflammation); finally, severity is determined, based on threshold levels of the phenotypic criteria [16].

The main purpose of this study was to assess the prevalence of malnutrition, as defined by the Global Leadership Initiative for Malnutrition (GLIM), in patients with COPD referred to pulmonary rehabilitation; and secondly, to explore associations of malnutrition according to GLIM consensus and its individual phenotypic criteria with mortality, hospital admissions (≥2), and hospital stay (>10 days) at 2-year follow-up.

## 2. Materials and Methods

### 2.1. Design

Post-hoc analysis of an on-going prospective cohort of COPD outpatients was carried out according to the Strengthening the Reporting of Observational Studies in Epidemiology recommendations [17].

### 2.2. Setting

The prospective cohort study was carried out in a pulmonary rehabilitation unit of a university hospital in Barcelona between June 2015 and December 2018.

### 2.3. Participants

Consecutive patients with a confirmed diagnosis of COPD referred to pulmonary rehabilitation were eligible for study participation. Patients with exacerbations and/or all-cause hospital admissions in the previous two months were excluded. At their first pulmonary rehabilitation visit, all patients are screened for malnutrition risk using the Mini-Nutritional Assessment Short Form (MNA-SF) [18]; a score ≤11 is considered as at-risk. For this study, the GLIM diagnostic criteria were also applied in at-risk patients [16], as described below.

### 2.4. Study Variables

Main outcome variables were malnutrition according to GLIM consensus, as well as 2-year mortality, hospital admissions, and length of stay, obtained from electronic medical records. For the purpose of analysis, dichotomous variables were established for hospital admissions (<2 vs. ≥2) and length of stay (<10 vs. ≥10 days) [6].

The GLIM consensus was assessed as follows:Phenotypic criteria (at least one of three characteristics): unintentional weight loss, defined as weight loss >5% within past 6 months or >10% beyond 6 months; low BMI (kg/m^2^), defined as <20 kg/m^2^ or <22 kg/m^2^ in participants younger and older than 70 years, respectively; and reduced muscle mass, identified as fat-free mass (FFM) <80% of the European reference values [19]. FFM was estimated by bioelectrical impedance analysis (Bodystat 1500, Bodystat Ltd., Isle of Man, British Isles) and calculated using sex-specific regression equations for patients with COPD [20]. Measures were expressed in kg and as a percentage of the European population reference values [20].Etiologic criteria (presence of at least one criterion): reduced food intake, based on response to the first item of the MNA-SF: “Has food intake declined over the past 3 months due to loss of appetite, digestive problems, chewing or swallowing difficulties?” [18]. In all the patients with moderate-to-very severe COPD, disease burden, and inflammation were considered present by definition.Malnutrition severity was assessed using the thresholds based on phenotypic criteria (weight loss, low BMI, and reduced muscle mass).

In addition to FFM, other parameters of body composition were measured: fat-mass (expressed in kg and as a percentage of the reference values) and water content (expressed in L and as a percentage of body weight); all these parameters were estimated by electrical impedance analysis. In addition to demographic characteristics (age, sex, and smoking history), other variables were collected:Severity of airflow obstruction according to Global Initiative for Obstructive Lung Disease criteria [21].Dyspnea evaluated with the modified scale of the Medical Research Council (mMRC) [22].Exercise capacity estimated with the distance travelled in the 6-minute walking test (6MWT) according to European standardized guidelines [23]. Patients were instructed to walk as far as possible in 6 min on a 30-meter walking course, while receiving the recommended encouragement. Arterial oxygen saturation and heart rate were measured by pulse oximetry and perceived dyspnea (Borg scale) was assessed before and after the trial.The multidimensional BODE (body mass index, obstruction, dyspnea, and exercise capacity) index was calculated. On this 10-point scale, higher scores indicate a higher risk of death [24].Respiratory function was tested with forced expiratory volume in the first second, forced vital capacity (FVC), total lung capacity, airway resistance, and carbon monoxide diffusing capacity (DL_CO_).

### 2.5. Ethics

National and international research ethics guidelines were followed, including the Deontological Code of Ethics and the Helsinki Declaration. Data were treated in accordance with the provisions of current laws in Spain and the General Data Protection Regulation in the European Union 2016/679 of the European Parliament and Council, dated 27 April 2016. The study was approved by the Comité d’Ètica de la Investigació amb Medicaments del Parc de Salut Mar (Research Ethics Committee for Medication of Parc de Salut Mar), and participants gave their written consent before inclusion in the study (Research ethics protocol number 2020/9395). All participants received individualized dietary advice and patients at risk or diagnosed of malnutrition received nutritional support from an expert nutritionist through the primary care system. At the end of the pulmonary rehabilitation follow-up, all patients referred to nutritional support remained under the care of a nutritionist.

### 2.6. Statistical Analysis

Clinical characteristics of the study participants were compared according to malnutrition status (as defined by the GLIM consensus) and two-year mortality. For these analyses, categorical variables were reported with absolute numbers and percentages, quantitative variables with the mean and standard deviation (SD) or median and percentiles as appropriate. The assumption of normality was analyzed with the Kolmogorov-Smirnov test. Chi-square test, Mann–Whitney U test, and Student *t* tests for paired and independent samples were used for the bivariate analysis as appropriate. For continuous variables, mean differences were indicated with 95% confidence intervals (95% CI). For each outcome, univariate and multivariate analysis applied malnutrition status (as defined by the GLIM consensus) as the main factor and was repeated separately for each of the GLIM consensus factors: unintentional weight loss, low age-related BMI (kg/m^2^), and low sex-related FFM. The relationship between GLIM criteria (overall and each component separately) and mortality at two years was checked with proportional hazard models. Results were expressed through hazard ratios (HR). The association of the GLIM criteria, also overall and each component separately, with hospital admissions (≥2) and hospital stays (≥10 days) was checked using binary logistic regression. Results for these analyses were expressed as odds ratios (OR). The inclusion of variables on multivariate models was performed balancing statistical criteria (those statistically significant variables on first bivariate analysis of malnutrition) and clinical criteria. Taking into account these criteria, two groups of adjusted models were created: first, adjusting for age and second adjusting also for pulmonary obstruction. A post-hoc power analysis was performed separately for each outcome and for both the crude analysis and for model 2, adjusted for age and obstruction severity. Data were processed using STATA 15.0 software (Stata Corp, College Station, TX, USA). All results were considered statistically significant at the 5% critical level.

## 3. Results

Of 200 consecutive patients referred to pulmonary rehabilitation in the study period, 27 (13.5%) had recently experienced a COPD exacerbation or required hospitalization and 6 (3%) patients declined to participate in the study; therefore, the final study population was 167 patients (80% men, aged 66.5 (SD 9.0) years). All participants were community-dwelling. More than 70% of them had severe or very severe airflow obstruction and most (83%) had been living with COPD for more than 5 years. Table 1 summarizes the baseline characteristics of participants.

Eighty-three (49%) patients were considered as at-risk of malnutrition (MNA-SF score ≤11), from which malnutrition was confirmed in 75 individuals, obtaining a prevalence of malnutrition according to the GLIM criteria of 45%. No significant differences in age, sex, or smoking history were observed between patients with and without malnutrition. Considering the variables commonly assessed previous to pulmonary rehabilitation, malnutrition was significantly associated with greater severity in the parameters of respiratory function (except FVC) and with body composition parameters: FFM, fat mass, and water content. Table 2 describes the main differences between COPD patients with and without malnutrition according to the GLIM criteria. The predicted distance in the 6MWT and the BODE index were significantly worse in patients with malnutrition. No significant differences between groups were found in the mMRC scale for dyspnea.

In the first year of follow-up, mortality was only 2.4% (*n* = 4), increasing to 12.4% (*n*= 21) in the second year. Of the 21 patients who died, 14 (66.6%) met the GLIM criteria for malnutrition in the assessment prior to beginning the rehabilitation program. As shown in Table 3, mortality was significantly associated with age, severity of symptoms (dyspnea, exercise capacity), body composition, BODE index, and lower FVC and DL_CO_ values.

A more detailed analysis by proportional hazard models is summarized in Table 4, which presents the associations of the GLIM diagnosis of malnutrition and each of its individual components with mortality, hospital admissions, and length of stay. Malnutrition according to GLIM criteria was associated with an increased risk of 2-year mortality (HR 2.8, 95% CI 0.9 to 8.0, *p* = 0.05). The analysis was repeated using each component of the GLIM phenotypic criteria. Mortality risk was almost 4 times greater for patients with low BMI (HR 3.7, 95% CI: 1.4 to 9.9, *p* = 0.009) and with low sex-specific FFM (HR 3.6, 95% CI: 1.3 to 9.9, *p* = 0.013). These associations remained significant after adjusting for age and for both age and lung function.

Malnutrition was associated with a nearly 3 times higher risk of ≥2 hospital admissions at 2-year follow-up (OR 2.9, 95% CI 1.4 to 6.0, *p* = 0.004), as well as a low MNA-SF score (OR 2.4, 95% CI 1.2 to 4.9, *p* = 0.015). Among the GLIM components, a low age-related BMI was associated with increased risk of hospital admissions (OR 2.7, 95% CI: 1.2 to 6.1, *p* = 0.018). All of these associations remained significant when adjusted for age, but were lost when also adjusted for lung function.

A hospital stay ≥10 days was associated with malnutrition (OR 2.6, 95% CI 1.2 to 5.5, *p* = 0.01) and with low MNA-SF scores (OR 2.4, 95% CI 1.2 to 5.2, *p* = 0.016). Among the individual GLIM components, a low age-related BMI was associated with almost 3 times higher risk of long-term hospital stay (OR 2.7, 95% CI: 1.2 to 6.2, *p* = 0.02). All of these associations remained significant after adjusting for age, but were lost when adjusted for age and obstruction severity. Unintentional weight loss and low FFM were associated with mortality but not with worse outcome among survivors.

Results from power calculation for the adjusted models (age and lung function) showed a 0.35 power for 2-year mortality, 0.40 for hospital admissions (≥2), and 0.25 for hospital stay ≥10 days.

## 4. Discussion

This study assessed the prevalence of malnutrition according to GLIM criteria in patients with stable COPD referred to pulmonary rehabilitation, and analyzed its association with prognostic variables (mortality and hospitalization) in a 2-year follow-up period. This analysis was previously done in our cohort, using the ESPEN consensus, in 2018 (*n* = 118, at the time of the study), showing a 24.6% prevalence of malnutrition and its association with 4 times greater mortality risk at 2-year follow-up [6]. About 2 years later, the cohort contained 200 patients and the observed prevalence of malnutrition according to GLIM was nearly doubled (45%).

In other studies, using the ESPEN criteria, the reported prevalence is consistent with our 2018 findings: 19.8% in rehabilitation patients with stable COPD [25] and 21% in hospitalized COPD patients [26]. Similarly, increased diagnoses of malnutrition when using the GLIM criteria have been reported in other populations. In a cohort of patients with systemic sclerosis, malnutrition was 17.9% according to ESPEN criteria and increased to 62.5% when the GLIM criteria were applied [27]. In a Canadian study in 18 hospitals (*n* = 1020), which found 45.2% malnutrition prevalence with the Subjective Global Assessment, the prevalence increased to 53.1% with the GLIM definition [28]. These increased results in some populations might be explained by the definition of malnutrition as “any disease burden or inflammatory condition”, accompanied by any one phenotypic criterion. The population with COPD frequently shows “reduced muscle mass” as phenotype, and so a large group of patients easily meet both criteria; however, an appropriate diagnosis and treatment of malnutrition may require consideration of other patient characteristics. For example, of the four COPD phenotypes proposed in the current Spanish guidelines [29], exacerbator patients with emphysema are more likely to have low BMI and low fat-free mass, compared to the other phenotypes: non-exacerbator, mixed COPD-asthma, and exacerbator with chronic bronchitis. In order to boost success of therapeutic interventions in COPD (including adherence to pulmonary rehabilitation), it is important to consider the individual patient’s phenotype, symptoms, and comorbidities [30,31].

The association of GLIM criteria and mortality is in line with results reported in previous studies. In a recent study based on hospitalized patients with hematological malignancy, the HR of one-year mortality is 2.39 (95% CI 1.36 to 4.20, *p* =  0.002) [32]; another study in advanced cancer patients shows a similar association between malnutrition according to the GLIM criteria and 6-month mortality (OR: 1.87. 95% CI 1.01 to 3.48, *p* = 0.047) [33]. Two out of three components of the GLIM criteria (low BMI and FFM) have been independently associated with an increased risk of mortality [6,26]. Weight loss alone, which is considered an indicator of health status in older people [34], was not associated with increased risk of mortality in our study. Available studies describe the capacity of the GLIM consensus to predict mortality, but there are scarce data about the association with morbidity.

Nutrition disorders (e.g., malnutrition) and nutrition-related conditions (e.g., sarcopenia) are common in patients with COPD, particularly in the oldest [35,36,37]. However, while malnutrition underlies an imbalance between energy intake and expenditure [15,16], and it may associate an inflammatory disease [16]; sarcopenia is a muscle disease rooted in adverse muscle changes [38], which may or may not be associated with another disease or inflammatory process [38]. Since decreased muscle mass is one of the foremost features of sarcopenia and malnutrition, they can overlap in some patients. A recent study comparing three nutritional screening tools with the GLIM criteria found that hospitalized older patients at high nutritional risk by the Malnutrition Universal Screening Tool were more likely to present with sarcopenia [39]. In our sample, the association among the two entities is clear: from 75 malnourished patients, 46 fulfill the EGSWOP2 criteria of sarcopenia. In addition, malnutrition is a strong predictor of sarcopenia and severe sarcopenia [40]. Defining nutritional disorders and malnutrition-related clinical conditions in specific populations should be a priority in future studies.

There is no doubt that malnutrition impacts on the prognosis of patients with COPD [6]. In our study, low age-related BMI and low sex-related FFM were the GLIM components associated with increased mortality at 2-year follow-up in the crude and adjusted models, whilst the GLIM criteria showed significant associations only in the crude analysis. The power estimation for hospital admissions and hospital stay was fairly good in the crude analysis (0.84 and 0.74, respectively), but power values for Model 2 were lower (less than 0.5). Power estimations for mortality were also low (in the crude and adjusted analyses). In view of these results, further studies with larger samples would be required before concluding the predictive value of the GLIM criteria.

Some limitations must be taken into consideration when interpreting our results. The study is a post-hoc analysis without sample size calculation because the cohort consisted of stable COPD patients referred to pulmonary rehabilitation; the post-hoc power calculations showed that this small sample size was a limitation to take into account. We would highlight a possible selection bias, since the patients preselected to rehabilitation programs should be able to perform the therapies and cannot be considered representative of the entire COPD population. Moreover, bioimpedance analysis is not the gold-standard tool to assess muscle mass, although it remains a low cost, easy, and available method of estimating fat and lean body mass for research and clinical use, according to international consensus [38]. Other limitations related to assessment and management of nutrition should also be considered. It would have been very useful to have information of objective markers of inflammation as the serum C-reactive protein (CRP) when applying the GLIM criteria. Patients in this cohort had stable COPD; therefore, many of them were being treated with corticosteroids and most had required oral corticosteroids in the previous year during exacerbations; no significant impact of this medication on nutritional status was observed. Unfortunately, data on other drugs with potential effect on nutritional status were not registered in the database. All patients considered at risk of malnutrition (MNA-SF score ≤11) were referred to a nutritionist who provided diet advice and nutritional supplements if required, but information on nutritional treatment and follow-up was not recorded in our database; this might constitute a bias to be considered. Finally, the relationship observed between malnutrition status and illness severity (significantly worse lung function) lost significance in analyses adjusted by age and lung function. All these limitations should be addressed in further prospective studies with larger samples including objective measures of inflammation, such as serum CRP, to establish the predictive value of the GLIM criteria.

## 5. Conclusions

The prevalence of malnutrition according to the GLIM criteria is high (45%) in patients with stable COPD referred to pulmonary rehabilitation, with an association with mortality and of hospitalizations in a 2-year follow-up. Future research is required to validate these findings in longitudinal studies according to patient’s phenotypic profiles.

## Figures and Tables

**Table 1 nutrients-13-00369-t001:** Baseline description of participants (*n* = 167).

	Total Sample (*n* = 167)
Age (years)	66.5 (SD 9.0)
Sex (men)	135 (80%)
Charlson index:	
1–3	132 (79%)
4–6	28 (17%)
7–9	7 (4%)
Dyspnea (mMRC scale)	2.11 (SD 0.9)
Six-minute walking distance (m)	396.7 (SD 113.4)
Six-minute walking distance (% pred)	80.3 (SD 21.3)
BODE index	3.1 (SD 2.1)
Body mass index (kg/m^2^)	26.9 (SD 6.4)
Weight loss	75 (45%)
Years living with COPD:	
≥5 years	138 (83 %)
≤5 years	29 (17%)
Smoking history:	
Smoker	63 (37.9%)
Past smoker	104 (62.1%)
Severity of airflow obstruction (according to GOLD):	
Moderate	43 (25.4%)
Severe	65 (38.5f%)
Very severe	42 (32.1%)
Bioimpedance body composition:	
Fat-free mass (Kg)	47.0 (SD 10.1)
Fat-free mass (% pred.)	87.7 (SD 16.1)
Fat mass (Kg)	26.8 (SD 12.5)
Fat mass (% pred.)	149.6 (SD 74.0)
Water (L)	38.4 (SD 8.2)
Water (% body weight)	57.3 (SD 48.5)

Abbreviations: mMRC: modified Medical Research Council; % pred.: percentage of predicted value; BODE: body mass index, airflow obstruction, dyspnea, and exercise; GOLD: Global Initiative for Chronic Obstructive Lung Disease; SD: standard deviation.

**Table 2 nutrients-13-00369-t002:** Clinical characteristics of the participants according to malnutrition as defined by the Global Leadership Initiative on Malnutrition (GLIM) consensus (*n* = 167).

	No Malnutrition	Malnutrition	Mean Differences	
	(*n* = 92)	(*n* = 75)	(95% CI)	*p*-Value
Age (years)	66.7 (SD 8.8)	66.3 (SD 9.5)	0.4 (−2.43 to 3.17)	0.796
Dyspnea (mMRC scale)	2.1 (SD 0.9)	2.2 (SD 1.0)	−0.7 (−0.4 to 0.2)	0.657
Respiratory function test:				
- %FEV1/FVC	49.0 (SD 12.8)	41.6 (SD 11.7)	1.9 (3.6 to 11.3)	**<0.001**
- FEV1 (% pred.)	42.0 (SD 14.2)	33.9 (SD 13.2)	8.1 (3.9 to 12.3)	**<0.001**
- FVC (% pred.)	66.1 (SD 14.7)	62.9 (SD 20.1)	3.2 (−2.2 to 8.6)	0.241
- TLC (% pred.)	100.7 (SD 20.0)	107.7 (SD 21.8)	−7.0 (−14 to 0.04)	0.051
- DLCO (% pred.)	49.9 (SD 17.5)	37.9 (16.2)	12.0 (6.3 to 17.8)	**<0.001**
Six-minute walking distance (m)	402.0 (SD 96.2)	391.5 (SD 133.2)	10.5 (−26.2 to 47.3)	0.572
Six-minute walking distance (% pred.)	84.7 (SD 20.4)	75.2 (SD 21.4)	9.5 (3.05 to 16.0)	**0.004**
BODE index	2.7 (SD 2.0)	3.4 (SD 2.1)	−0.7 (−1.4 to −0.09)	**0.023**
Body mass index (kg/m^2^)	30.7 (SD 5.4)	22.4 (SD 4.4)	8.3 (6.8 to 9.8)	**<0.001**
Bioimpedance body composition:				
- Fat-free mass (kg)	52.2 (SD 9.8)	41.7 (SD 7.3)	10.5 (7.7 to 13.4)	**<0.001**
- Fat-free mass (% pred.)	97.2 (SD 14.2)	78.0 (SD 11.5)	19.2 (14.9 to 23.5)	**<0.001**
- Fat mass (kg)	32.2 (SD 11.6)	21.3 (SD 10.9)	10.9 (7.2 to 14.6)	**<0.001**
- Fat mass (% pred.)	179.1 (SD 71.2)	119.8 (SD 64.6)	59.3 (36.8 to 81.8)	**<0.001**
- Water (L)	42.2 (SD 8.3)	34.6 (SD 6.2)	7.5 (5.11 to 10.0)	**<0.001**
- Water (% body weight)	58.2 (SD 67.9)	56.3 (SD 8.1)	1.8 (−14.3 to 18.0)	0.825

Abbreviations: BODE: body mass index, obstruction, dyspnea, and exercise capacity; DLCO: carbon monoxide diffusing capacity; FEV1: forced expiratory volume in the first second; FVC: forced vital capacity; TLC: total lung capacity; % pred.: percentage of predicted value; mMRC: modified Medical Research Council. Significant HR (*p* < 0.05) indicated in bold.

**Table 3 nutrients-13-00369-t003:** Main clinical differences between survivors and non-survivors at 2-year follow-up.

	Survivors	Non-Survivors	Mean Differences	
	(*n* = 152)	(*n* = 15)	(95% CI)	*p*-Value
Age (years)	65.9 (SD 9.0)	72.7 (SD 8.1)	−6.8 (−11.5 to −2.0)	**0.005**
Dyspnea (mMRC scale)	2.1 (SD 0.9)	2.6 (SD 0.8)	−0.5 (0.09 to −0.02)	**0.042**
Respiratory function test:				
- %FEV1/FVC	45.8 (SD 12.4)	43.7 (SD 16.8)	2.1 (−4.8 to 9.0)	0.548
- FEV1 (% pred.)	38.8 (SD 13.9)	33.1 (SD 17.4)	5.7 (−1.9 to 13.3)	0.137
- FVC (% pred.)	65.4 (SD 17.4)	56.5 (SD 13.7)	8.9 (0.3 to 18.1)	**0.057**
- TLC (% pred.)	104.3 (SD 21.1)	98.4 (SD 19.3)	5.9 (−7.1 to 18.9)	0.37
- DLCO (% pred.)	45.9 (SD 17.7)	31.3 (SD 13.4)	14.6 (3.7 to 25.4)	**0.009**
Six-minute walking distance (m)	407.9 (SD 105.51)	283.5 (SD 131.1)	124.5 (66.8 to 182.2)	**<0.001**
BODE index	2.9 (SD 2.0)	4.3 (SD 2.0)	−14 (−2.4 to −0.32)	**0.011**
Body mass index (kg/m^2^)	27.4 (SD 6.4)	22.4 (SD 4.5)	5 (1.6 to 8.3)	**0.004**
Bioimpedance body composition:				
- Fat-free mass (kg)	47.8 (SD 10.3)	40.0 (SD 4.9)	7.2 (4.1 to 10.4)	**<0.001**
- Fat-free mass (% pred.)	89.0 (SD 16.3)	76.7 (SD 9.3)	12.3 (3.8 to 20.8)	**0.005**
- Fat mass (kg)	27.4 (SD 12.7)	20.0 (SD 8.4)	7.5 (0.9 to 14.1)	**0.027**
- Fat mass (% pred.)	154.8 (SD 75.5)	105.6 (SD 39.6)	49.2 (24.2 to 74.2)	**<0.001**
- Water (L)	39.0 (SD 8.4)	33.7 (SD 4.3)	0.3 (2.6 to 8.6)	**<0.001**
- Water (% body weight)	57.3 (SD 51.2)	57.0 (SD 10.3)	0.3 (−6.0 to 26.5)	0.984

Abbreviations: BODE: body mass index, obstruction, dyspnea, and exercise capacity; DLCO: carbon monoxide diffusing capacity; FEV1: forced expiratory volume in the first second; FVC: forced vital capacity; TLC: total lung capacity; **%** pred.: percentage of predicted value; mMRC: modified Medical Research Council. Significant HR (*p* < 0.05) indicated in bold.

**Table 4 nutrients-13-00369-t004:** Hazard ratios of mortality and odds ratios of admissions and length of stay in 2-year follow-up according to malnutrition as defined by the GLIM consensus and separately by its specific phenotypic criteria. Crude analysis and adjusted results (*n* = 167).

	Crude Analysis			Model 1 Adjusted for Age			Model 2 Adjusted for Age and Obstruction Severity		
	**HR**	**CI 95%**	***p***	**HR**	**CI 95%**	***p***	**HR**	**CI 95%**	***p***
**Mortality at 2 years: 16 cases**									
Malnutrition according to GLIM	2.8	0.9 to 8.0	**0.05**	2.8	1.0 to 8.1	**0.05**	2.27	0.8 to 6.8	0.140
Unintentional weight loss	2.2	0.5 to 9.8	0.29	2.0	0.4 to 8.7	0.38	1.75	0.4 to 7.9	0.46
Low age-related BMI (kg/m^2^)	3.7	1.4 to 9.9	**0.009**	3.8	1.4 to 10.1	**0.009**	3.13	1.1 to 9.0	**0.034**
Low sex-related FFM	3.6	1.3 to 9.9	**0.013**	3.6	1.3 to 9.8	**0.014**	3.18	1.1 to 8.8	**0.026**
**Hospital admissions (≥2): 42 cases**	**OR**	**CI 95%**	***p***	**OR**	**CI 95%**	***p***	**OR**	**CI 95%**	***p***
Malnutrition according to GLIM	2.9	1.4 to 6.0	**0.004**	2.9	1.4 to 6.0	**0.004**	1.89	0.9 to 4.1	0.116
Unintentional weight loss	1.1	0.3 to 4.4	0.87	1.1	0.3 to 4.6	0.84	1.08	0.2 to 5.0	0.91
Low age-related BMI (kg/m^2^)	2.7	1.2 to 6.1	**0.018**	2.7	1.2 to 6.1	**0.019**	1.70	0.7 to 4.2	0.24
Low sex-related FFM	1.3	0.6 to 2.7	0.54	1.3	0.6 to 2.7	0.54	0.9	0.4 to 2.1	0.84
**Hospital stay (>10 days): 38 cases**									
Malnutrition according to GLIM	2.6	1.2 to 5.5	**0.01**	2.6	1.2 to 5.6	**0.01**	1.65	0.7 to 3.8	0.23
Unintentional weight loss	0.7	0.2 to 3.6	0.70	0.7	0.2 to 3.6	0.71	0.64	0.1 to 3.6	0.61
Low age-related BMI (kg/m^2^)	2.7	1.2 to 6.2	**0.02**	2.7	1.2 to 6.2	**0.02**	1.67	0.6 to 4.2	0.27
Low sex-related FFM	1.0	0.4 to 2.2	0.94	1.0	0.4 to 2.2	0.94	0.65	0.3 to 1.6	0.35

Abbreviations: BMI: body mass index; CI: confidence interval; GLIM: Global Leadership Initiative on Malnutrition; FFM: fat-free mass; HR: hazard ratio; OR: odds ratio; Model 1: adjustment for age; Model 2: adjustment for age and obstruction severity; significant HR (*p* < 0.05) indicated in bold.

## Data Availability

Further supporting data is available from the authors on request.

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
