# Peer review of "Malnutrition According to GLIM Criteria Is Associated with Mortality and Hospitalizations in Rehabilitation Patients with Stable Chronic Obstructive Pulmonary Disease"

_nutrients, 2021, doi:10.3390/nu13020369_

Round 1
Reviewer 1 Report
The authors report the results of a post-hoc analysis of the prevalence of malnutrition in a COPD cohort of patients who were assessed by the GLIM criteria.
Comments:
Globally this is a valuable study assessing the value of the recently proposed GLIM criteria in a specific population.
The first limitation is a post-hoc analysis.
1 The authors describe they started the evaluation by using the MNA-SF score;but they do not detail what was the percentage of the population that was considered to be 'at risk of malnutrition' and that was subsequently assessed with the GLIM criteria.
2 For the patients who were 'at risk' the GLIM criteria were used;could you confirm that was done only at the first evaluation ?The authors report that 45 % of the cohort of patients were malnourished according to the GLIM crtiteria;they also used this criteria for assessing the prognosis of the population in terms of mortality and hospitalization;but we have no data on the nutritional intervention that was initiated after this initial evaluation;a nutritional intervention may obviously interfer with the prognosis.
3 The authors consider that all the patients met the criteria of ' disease burden and inflammation';this is a debatable problem of the GLIM criteria;should we propose to use a more objective value as the serum CRP.?In this group of patients the degree of inflammation may be fluctuating along with the time.
4 Taking into account that all the patients had a positive etiologic factor it seems that weight loss was the major criteria to define malnutrition?
What is the adding value of measuring the other criteria ?
Considering the prognosis a low-BMI seems to be more valuable.
5 In the design of the study the cutt-offs for defining the severity of malnutrition were not detailed.
Did you find any value of using the severity of malnutrition ?
The authors should discussed the adding value of the GLIM criteria in comparison with other criteria such as SGA.
In this report there is no data about a nutritional intervention that may impact the prognosis;do you really consider that the GLIM criteria may provide a prognosis in this population ?.
Reviewer 2 Report
The paper by Vanesa Dàvaros-Yerovi and co-workers describes the results of a post-hoc analysis from a prospective cohort of patients affected by COPD undergoing rehabilitation. They found a 45% prevalence of malnutrition according to GLIM criteria. Furthermore, malnutrition in this cohort was associated with 2-year mortality and hospitalization risk.
The topic of this investigation is of interest, the manuscript is well written. Nevertheless, the Authors need to address the following concerns:
- prevalence of malnutrition in this cohort of COPD patients is very high as compared with previous studies (PMID 30349235, 29413494); this may be related to exclusion criteria, which are apparently not strict in this study. Did the Authors include patients affected by other co-morbidities such as cancer, neuromuscular disease, dysphagia, diabetes, cognitive decline, liver/kidney failure? If not, this information needs to be included since it could affect the overall results of the study. Furthermore, information on pharmacotherapy should be provided (did any patient take drugs potentially affecting nutritional status, such as corticosteroids, vitamin D, L-tyroxin...?);
- the Authors should study factors associated with malnutrition in their cohort (smoking status, years with COPD, GOLD stage, exacerbation status...?)
- it is interesting to note that both fat-free and fat mass were lower in the malnutrition group as compared to no-malnutrition. The Authors should discuss the coexistence of COPD and malnutrition with sarcopenia (PMID 29413494, 32560480)
Minor concerns:
- please check "lenghts of stay" on line 99;
- please check for percentage of smoker subjects in table 1.
Round 2
Reviewer 1 Report
The authors did consider the comments, make changes and underline the limitations of this study.
Globally this paper may contribute for the validation of the GLIM criteria.
Reviewer 2 Report
The Authors replied to the concerns raised.